

# Effect of grain-sorting waves on alternate bar dynamics: Implications of the breakdown of the hydrograph boundary layer

Soichi Tanabe[1], Toshiki Iwasaki[2]

[1]Graduate School of Engineering, Hokkaido University, Sapporo, 060-8628, Japan
[2]Faculty of Engineering, Hokkaido University, Sapporo, 060-8628, Japan

*Correspondence to*: Soichi Tanabe (okushishiku@eis.hokudai.ac.jp)

**Abstract.** Understanding the morphological responses of gravel-bed rivers to changes in external forces (e.g. water and sediment supply conditions) is a critical concern in river science and engineering. However, this remains a challenging issue because river responses are highly dependent on the distance from the source point where such environmental changes occur.

Here, we focus on the short-term effects of flood-scale non-equilibrium sediment supply on the downstream alternate bar dynamics in poorly sorted gravel-bed rivers using a numerical morphodynamic model. Specifically, we perform a two-dimensional morphodynamic calculation using iRIC-Nays2DH in a straight channel under repeated cycles of an unsteady water hydrograph and a constant supply of poorly sorted sediment. Under the well-sorted sediment cases, the upstream non-equilibrium sediment supply can affect only a limited distance from the upstream end (i.e. the hydrograph boundary layer).

However, the inclusion of a poorly sorted sediment disrupts this concept, leading to the migration of low-amplitude bedload sheets far downstream. In this context, the upstream water and sediment boundary conditions may affect the far-downstream river dynamics through the migration of bedload sheets. The numerical results show that the migration of bedload sheets and the associated fine sediment transport greatly affect the alternate bar dynamics and change their texture. However, this effect of bedload sheets on bars cannot propagate across the entire channel and disappears completely in the alternate bars located

further downstream. These results suggest that the upstream non-equilibrium sediment supply condition in poorly sorted sediment has a non-negligible role in downstream alternate bar dynamics even far from the sediment feed point. However, this effect becomes negligible in the further downstream reaches as long as active and larger morphological changes, such as alternate bars, greatly disperse the bedload sheets.

## 1 Introduction

Continuous and/or episodic changes in external forces caused by various factors (e.g. climate change [Trenberth, 2011], coseismic mountain collapse [Schuerch et al., 2006], installation and removal of dams [Fields et al., 2021], chute cut-off [Zinger et al., 2011], post-wildfire erosion [Benda et al., 2003], and sediment augmentation [Mörtl and De Cesare, 2021]) are critical in controlling the dynamics of rivers. The hydrograph and sediment supply, which are the most common external factors affected by these changes, have a significant impact on the channel geometry [Venditti et al., 2019], riverbed





composition [Nelson et al., 2009], and vegetation [Erskine et al., 1999]. These river responses are also dependent on the dominant bed material [Gaeuman et al., 2005] and sediment transport mode [Gunsolus and Binns, 2017].

Gravel-bed rivers composed of poorly sorted sediments generally have clear three-dimensional bedform structures, such as fluvial bars. The effects of the hydrograph and sediment supply on fluvial bars have been investigated through field surveys, laboratory experiments, and numerical calculations, demonstrating their significant impact on bar dynamics. For example,

constant water and equilibrium sediment supply conditions result in a regular pattern of free bars in terms of their shape characteristics (i.e. mode, wavelength, and bar height) [e.g. Colombini et al., 1987]. Meanwhile, a non-equilibrium sediment supply provides a spatially varying bar shape and a corresponding surface texture pattern, regardless of the upstream water discharge conditions [Lisle and Hilton, 1999; Nelson et al., 2015; Morgan and Nelson, 2021]. A reduction in the sediment supply suppresses the mobility of the riverbed material, resulting in the formation of coarse patches [Dietrich et al., 1989],

coarsening of the corridor [Lisle et al., 1993], and dissipation of the bar structure [Venditti et al., 2012]. However, an increase in sediment supply generally causes greater mobility of the sediment and associated bed evolution, leading to the formation of shorter ephemeral bars with high migration rates [Podolak and Wilcock, 2013; Bankert and Nelson, 2018; Nelson and Morgan, 2018]. Furthermore, the response of fluvial bars under unsteady flow differs from that under steady flow [Tubino, 1991; Huang et al., 2023]. In addition, some specific hydrograph characteristics cause unique riverbed forms [e.g. Waters and Curran, 2015]

and grain size compositions [e.g. Hassan and Church, 2001] in the rising and falling limbs of a single hydrograph, thus contributing to the non-linear hysteresis of sediment transport [Gunsolus and Binns, 2017]. This hysteresis varies according to the hydrograph shape [Bombar et al., 2011], duration [Hassan et al., 2006], and magnitude [Lee et al., 2004]. These studies indicated that both sediment supply and the hydrograph are critical components in controlling sediment transport and thus the responses of bars composed of poorly sorted sediment, strongly suggesting the importance of understanding upstream water

and sediment supply conditions on fluvial river morphodynamics.

One of the difficulties in understanding the relationship between sediment supply conditions and the corresponding riverbed grain size responses is that these responses are dependent on the distance from the source point of sediment supply/reduction, particularly in bedload-dominated river reaches (i.e. gravel-bed rivers composed of poorly sorted sediment). Even in rivers where suspended transport is dominant, the distance from the source point can be an important factor [An et al.,

2018]; however, because suspended material has a longer transport distance than bedload material, the channel may respond farther downstream from the sediment feed point [e.g. Andrews, 1986]. In the case of bedload-dominated rivers, sediment supply/reduction gradually affects the downstream bed, and the grain size changes over a much longer timescale. We provide a few field-scale examples of such bedload-dominated cases. Fields et al. (2021) investigated the temporal transition of the channel geometry for several years after dam removal. They found that channel geometries changed significantly in the vicinity

of the removed dam, i.e. channel incision in the upstream reach and bed aggradation with channel widening in the downstream reach. However, there was little change in the channel geometry a few hundred meters downstream, suggesting an effective length scale of the sediment source on the downstream morphodynamic changes. A similar example can be found in the debate regarding the cause of the Mississippi Delta retreat. The Mississippi Delta retreat has been understood to be the result of a





reduction in sand supply due to dam construction [Blum and Roberts, 2009]. However, Nittrouer and Viparelli (2014)

suggested a more direct cause using a one-dimensional morphodynamic model: the effects of the reduction in sand supply have not reached the delta area far from the dams. These examples suggest the importance of understanding the extent to which the effects of external forces, such as changes in sediment supply conditions, on downstream river morphodynamics can propagate in time and space. This is a challenging task, particularly in natural streams, because many other factors such as bending [Buraas et al., 2014], the original riverbed composition [Gaeuman et al., 2005], and vegetation [White et al., 2023]

control the channel geometry.

Wong and Parker (2006) clearly quantified the length of river reach that is strongly affected by non-equilibrium sediment supply within a single hydrograph using simplified experiments. They intentionally set the upstream boundary condition as a non-equilibrium sediment supply condition using a cycled triangular hydrograph and a constant sediment supply. This boundary condition led to cyclic behaviour of bed aggradation at low discharge owing to the oversupply of sediment to the

capacity and degradation at high discharge caused by the limited supply condition at the upstream end (Fig. 1). However, this bed fluctuation propagated only limited length downstream, defined as the "hydrograph boundary layer" (referred to as HBL hereafter). Using well-sorted sediments, as in their experiment, the HBL represents a typical length scale of the effect of sediment source/reduction on the downstream bed evolution of gravel-bed rivers within a single flood event.

In contrast to the well-sorted sediment case, in a poorly sorted sediment bed, grain-sorting waves are generally formed

owing to a non-equilibrium sediment supply, such as mountain collapse, sediment augmentation [An et al., 2017; Venditti et al., 2010a, b], and repeated sediment release from the dam bypass tunnel [Facchini et al., 2024]. An et al. (2017) performed one-dimensional morphodynamic calculations under conditions similar to those of Wong and Parker (2006), with the exception that they targeted poorly sorted sediments. They observed similar bed fluctuation characteristics as in the well-sorted sediment case, i.e. the HBL-like, with a limited propagation distance of bed fluctuation due to the non-equilibrium sediment supply.

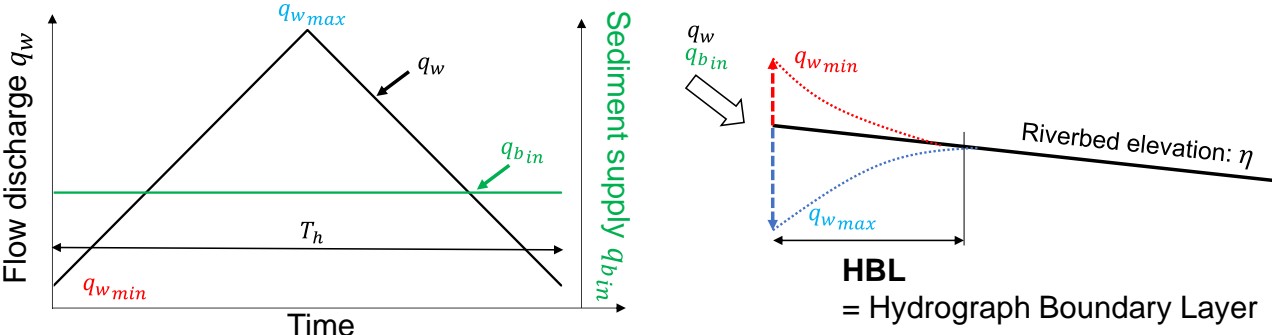

Fig. 1 The concept of hydrograph boundary layer (HBL). $q_w$ is the flow discharge, $q_{w_{max}}$ and $q_{w_{min}}$ are the maximum flow discharge and the minimum flow discharge, respectively, $q_{b_{in}}$ is the sediment supply from upstream end, $T_h$ is the duration of one single hydrograph, and $\eta$ is the riverbed elevation.



However, they also showed that an advection–diffusion-type grain-sorting wave could migrate far downstream from the upstream end, suggesting a breakdown of the HBL concept in poorly sorted sediment. This grain-sorting wave had similar characteristics to the low-amplitude and long-wavelength bedload sheet found in gravel-bed rivers, with a grain-scale tip containing coarse particles, behind which is filled with fine particle interstices of coarse particles [e.g. Whiting et al., 1988]. Iseya and Ikeda (1987) and Whiting et al. (1988) reported two distinct reaches of bedload sheets: 1) a "matrix-filled gravel

layer", where fine particles are filled in interstices between coarse particles, and 2) an "open-work gravel layer", which is starved for fine particles. The migrating mechanism of bedload sheets is as follows: 1) in the reach with an open-work gravel layer, coarse particles from upstream are deposited until reaching a critical slope that allows sediment to move downstream; 2) after reaching the critical slope and stabilising the riverbed surface, the interstices of coarse particles are completely filled with fine particles, producing a matrix-filled gravel layer; 3) the fill of fine particles creates a smooth surface, and coarse

particles are transported; and 4) coarse particles are separated from the fine particles because of the difference in step length, and only coarse particles are transported downstream from the reach with the open-work gravel layer. As described in 3) above, bedload sheets smooth the surface and reduce the internal friction angle [e.g. Wilcock, 1998; Wilcock et al., 2001], thus increasing the total sediment transport rate associated with the bedload sheet migration [Whiting et al., 1988; Nelson et al., 2009]. In summary, the migration of bedload sheets causes ephemeral non-equilibrium sediment transport far downstream

from the sediment source/reduction point, indicating a possible effect of upstream sediment conditions on the far-downstream bed and grain size dynamics [An et al., 2017]. Dai et al. (2021) indicated that as the concept of the HBL suggests, the alternate bars downstream of the HBL are not affected by upstream non-equilibrium conditions in the case of uniform-sized sediment. However, in the case of poorly sorted sediment, the bars that develop far downstream from the upstream sediment supply/reduction point are expected to be affected by bedload sheets. Because bars composed of heterogeneous sediments are

more unstable under external forces than those composed of homogeneous sediments [Lanzoni and Tubino, 1999], even low-amplitude grain-sorting waves may have a non-negligible effect on the downstream bar dynamics.

    Only a few studies have addressed the interactions between sediment waves and bars. For example, Lisle et al. (1997) conducted a field-scale experiment (with a flume length of 160 m) on the dynamics of sediment pulses over migrating alternate bars using well-sorted sediment. The sediment pulse was a combination small-advection and diffusion wave; the effect did not

propagate to the downstream end of the flume. This implies that the sediment wave generated under well-sorted sediment has a limited effect on the downstream bed morphodynamics. This is consistent with the concept of the HBL; however, a poorly sorted sediment case may show different downstream sediment behaviours and morphodynamics. Humphries et al. (2012) experimentally investigated the sediment pulse dynamics in a channel with a riffle-pool sequence intentionally created to mimic a natural bedform (channel length of 28 m). The effect of the sediment pulse propagated from the pulse feed point to

the downstream end of the channel, suggesting that the sediment pulse could affect a significantly longer distance in the channel. This type of experiment provides important insights into the effective spatiotemporal scale of sediment pulses to the downstream riverbed; however, the limited length of the channel can be a critical concern in demonstrating sediment wave migration and associated morphodynamics, even in field-scale experiments. Recent advances in numerical morphodynamic



models provide a sufficient capability to reproduce complex morphodynamic components such as fluvial bar dynamics [e.g.
Shimizu et al., 2020] with no limitation of the spatial scale, such as the channel length; therefore, these models have been a
powerful tool for understanding large-scale sediment transport and morphodynamics, including the breakdown of the HBL
and its implications for bedload sheet formation [e.g. An et al., 2017].

In this study, a numerical morphodynamic model, iRIC-Nay2DH was employed to investigate the behaviour of rivers
with alternate bars and bedload sheets composed of poorly sorted sediment subjected to cyclic triangular hydrographs and a
constant sediment supply. More specifically, we focused on 1) the effect of bedload sheets on alternate bars and 2) the
behaviour of bedload sheets inside the bars. Our study differs from the previous study by An et al. (2017) in that 1) our study
is extended to two-dimensional calculations and considers three-dimensional riverbed morphology, i.e. alternate bars, and 2)
our hydrograph targets one short-scale (i.e. flash flood) single flood repetition, whereas An et al. (2017) explored the repetition
of long-term changes in the flow regime. We simplified the channel geometry and upstream conditions (i.e. a straight channel
with a wide rectangular cross-section, symmetric triangular hydrograph, and constant sediment supply) to provide a simple
representation of the morphodynamic responses that can occur when the sediment supply volume and sediment transport
capacity do not match under unsteady flow conditions within a single hydrograph.

## 2 Numerical model

### 2.1 Model formulation

In this study, we employed the Nays2DH model [Shimizu et al., 2014] implemented in iRIC software [Nelson et al., 2016]
as a computational morphodynamic model to simulate fluvial bars with poorly sorted sediment, such as in typical gravel-bed
rivers, under non-equilibrium sediment supply conditions caused by unsteady-flow discharge with a constant sediment supply.
This model has been applied to various morphodynamic phenomena in rivers, and it can sufficiently capture the basic physics
of riverbed evolution under mixed-sized sediment conditions [e.g. Iwasaki et al., 2011; Harada et al., 2019; Harada and
Egashira, 2023]. Note that we implemented some functions for the sediment mixture module in the original iRIC-Nays2DH
(i.e. calculation of the geometric mean diameter, spatiotemporal variation in Manning's roughness coefficient due to surface
grain size changes, the bedload transport relation proposed by Wilcock and Crowe (2003), and boundary conditions for
sediment recirculation).

The flow model is an unsteady two-dimensional shallow-water model. The governing equations for this model are written
for a generalised coordinate system. For simplicity, we describe these in the Cartesian coordinate system herein as follows:

$$\frac{\partial h}{\partial t} + \frac{\partial uh}{\partial x} + \frac{\partial vh}{\partial y} = 0, \qquad (1)$$

$$\frac{\partial u}{\partial t} + u\frac{\partial u}{\partial x} + v\frac{\partial u}{\partial y} = -g\left(\frac{\partial h}{\partial x} + \frac{\partial \eta}{\partial x}\right) - \frac{gn_m{}^2 uV}{h^{\frac{4}{3}}}, \qquad (2)$$



$$\frac{\partial v}{\partial t} + u\frac{\partial v}{\partial x} + v\frac{\partial v}{\partial y} = -g\left(\frac{\partial h}{\partial y} + \frac{\partial \eta}{\partial y}\right) - \frac{gn_m{}^2 vV}{h^{\frac{4}{3}}}, \qquad (3)$$

where $x$ and $y$ are the downstream and transverse coordinates, respectively, $t$ is the time, $h$ is the water depth, $u$ and $v$ are the depth-averaged flow velocity components in the $x$ and $y$ directions, respectively, $V$ is the composite velocity ($= \sqrt{u^2 + v^2}$), $\eta$ is the riverbed elevation, $g$ is the gravitational acceleration, and $n_m$ is Manning's coefficient. This coefficient is updated as the riverbed texture changes according to the Manning–Strickler roughness formula as follows:

$$n_m = \frac{k_s{}^{\frac{1}{6}}}{7.66 g^{\frac{1}{2}}}, \qquad (4)$$

$$k_s = 2.5 d_g, \qquad (5)$$

where $k_s$ is the roughness height, and $d_g$ is the geometric mean diameter.

We use an active layer formulation [Hirano, 1971] to simulate the evolution of the riverbed and the surface grain size distribution in a poorly sorted sediment riverbed. Assuming that the bed porosity and active layer thickness are constant, the riverbed elevation and surface grain size distribution are updated as follows:

$$\frac{\partial \eta}{\partial t} = -\frac{1}{(1-\lambda)}\left(\frac{\partial q_B{}^x}{\partial x} + \frac{\partial q_B{}^y}{\partial y}\right), \qquad (6)$$

$$\frac{\partial F_{ai}}{\partial t} = -\frac{1}{L_a(1-\lambda)}\left\{F_{ai}\frac{\partial \eta}{\partial t} + \left(\frac{\partial q_{Bi}{}^x}{\partial x} + \frac{\partial q_{Bi}{}^y}{\partial y}\right)\right\}, \qquad (7)$$

where $q_B{}^x$ and $q_B{}^y$ are the bedload transport rate per unit width in the $x$ and $y$ directions, respectively, the subscript $i$ indicates physical quantities of the $i$th grain size class, $F_{ai}$ is the volumetric fraction of the $i$th grain size class in the active layer ($\sum F_{ai} = 1$), $\lambda$ is the porosity of the riverbed, and $L_a$ is the active layer thickness, which affects the sensitivity of the riverbed evolution in the numerical calculation. In general, the active layer thickness is evaluated as a linear function of the representative diameter, e.g. the geometric mean diameter, $d_g$. In this study, we set $L_a$ to twice $d_g$ as follows:

$$L_a = 2d_g. \qquad (8)$$

With respect to $F_{ai}$, the grain size fraction in the active layer is adopted when the riverbed aggrades, and the grain size fraction in the substrate is adopted when the riverbed degrades, as described in detail below.

The morphodynamic features considered in this study are characterised by the cyclic behaviour of bed aggradation/degradation caused by a non-equilibrium sediment supply at the upstream end and the migration of free alternate bars in the straight channel. Both components may lead to a distinct grain-sorting layer in the riverbed. In addition, the surface texture and bar structure exhibit hysteresis under unsteady flow [e.g. Hassan and Church, 2001; Mao, 2012; Wang et al., 2019]. To capture this stratigraphic record, Nays2DH stores the grain size distribution at the surface and inside the bed using a three-layer approach, i.e. an active surface layer, a deposition layer in the bed, and a transition layer in between [Ashida et al., 1990]. The substrate bed layer is divided into a transition layer and several deposition layers. The transition layer is the intermediate layer between the active and deposition layers, meaning that it transitions from the deposition layer to the active layer, or vice



versa.

In this study, we focused on the morphodynamics of poorly sorted gravel-bed rivers and considered bedload transport as the only mode of sediment transport. For this purpose, we employed the bedload transport relation proposed by Wilcock and Crowe (2003), which is applicable to a wide range of grain size distributions.

$$q_{Bi}{}^s = \frac{W_i F_{ai} u_*^3}{Rg}, \qquad (9)$$

where the superscript $s$ is the local streamwise direction coordinate, $R$ is the submerged specific gravity of the sediment, $u_*$ is the shear velocity, and $W_i$ is the dimensionless bedload transport rate calculated from the following equation:

$$W_i = \begin{cases} 0.002\phi_i^{7.5} & \phi_i < 1.35 \\ 14\left(1 - \dfrac{0.894}{\phi_i}\right)^{4.5} & \phi_i \geq 1.35 \end{cases}, \qquad (10)$$

where the dimensionless parameter $\phi_i$ is defined as the ratio of the bed shear stress, $\tau_b$, to the reference shear stress for the $i$th grain size class, $\tau_{ri}$.

$$\phi_i = \frac{\tau_b}{\tau_{ri}} \qquad (11)$$

The bed shear stress, $\tau_b$, is evaluated as follows:

$$\tau_b = \rho g h i_e = \frac{\rho g n_m^2 V^2}{h^{\frac{1}{3}}}, \qquad (12)$$

where $\rho$ is the water density, and $i_e$ is the energy gradient. The reference shear stress, $\tau_{ri}$, is given as follows:

$$\tau_{ri} = \left(\frac{d_i}{d_g}\right)^b \tau_{rg}{}^* R\rho g d_g, \qquad (13)$$

where $\tau_{rg}{}^*$ is the dimensionless reference shear stress for the geometric mean size calculated as a function of the fraction of sand in the active layer, $F_s$, as follows:

$$\tau_{rg}{}^* = 0.021 + 0.015 \exp(-20F_s). \qquad (14)$$

The exponent $b$ characterises the hiding effect among different grain sizes and is computed as follows:

$$b = \frac{0.67}{1 + \exp\left(1.5 - \dfrac{d_i}{d_g}\right)}. \qquad (15)$$

The bedload transport rate for the transverse direction is calculated as follows:

$$q_{Bi}{}^n = q_{Bi}{}^s \left(\frac{v_{cb}{}^n}{V_{cb}} - \sqrt{\frac{\tau_{*ri}}{\mu_s \mu_k \tau_{*i}}} \frac{\partial \eta}{\partial n}\right), \qquad (16)$$

where $n$ is the transverse coordinate, $\tau_{*i}$ is the dimensionless shear stress of the $i$th grain size class, $\tau_{*ri}$ is the dimensionless reference shear stress of the $i$th grain size class, $v_{cb}{}^n$ is the flow velocity near the riverbed in the $n$ direction, $V_{cb}$ is the composite velocity near the riverbed, and $\mu_s$ and $\mu_k$ are the static and dynamic friction coefficients, respectively. The bedload





transcript vector in Cartesian coordinates can be calculated from the bedload vector in local streamwise coordinates $(q_{Bi}{}^s, q_{Bi}{}^n)$ based on the depth-averaged flow vector, as follows [e.g. Iwasaki et al., 2016]:

$$q_{Bi}{}^x = \frac{\partial x}{\partial s} q_{Bi}{}^s + \frac{\partial x}{\partial n} q_{Bi}{}^n = \cos \theta_s\, q_{Bi}{}^s - \sin \theta_s\, q_{Bi}{}^n = \frac{u}{V} q_{Bi}{}^s - \frac{v}{V} q_{Bi}{}^n, \qquad (17)$$

$$q_{Bi}{}^y = \frac{\partial y}{\partial s} q_{Bi}{}^s + \frac{\partial y}{\partial n} q_{Bi}{}^n = \sin \theta_s\, q_{Bi}{}^s + \cos \theta_s\, q_{Bi}{}^n = \frac{v}{V} q_{Bi}{}^s + \frac{u}{V} q_{Bi}{}^n, \qquad (18)$$

where $\theta_s$ is the angle of the depth-averaged flow along the $x$ axis.

## 2.2 Model validation

In this section, we validate the iRIC-Nays2DH model for application to the morphodynamics of fluvial bars in poorly sorted sediment by reproducing the flume experiments of Nelson et al. (2010). Their experiments aimed to explore the bed surface topography and texture over a gravel bed of quasi-steady alternate bars. This flume was located at the St. Anthony Falls Laboratory at the University of Minnesota in Minneapolis. The flume width was 2.75 m, channel length was 55 m, and slope was 0.013. Flow discharge was held constant at $0.4 \pm 0.02$ m³/s for approximately 20 h. The sediment used in the experiment was poorly sorted gravel of 2–45 mm in diameter, with a geometric mean diameter of 11.2 mm. The sediment was recirculated. A block was installed to cover one-third of the flume entrance to trigger the formation and development of alternate bars.

During the experiment, the water surface elevation, local flow velocity, and sediment runoff at the downstream end were recorded, and photographs were captured to analyse the surface texture, as explained below. After the experiment, they investigated the high-resolution riverbed elevation and automated surface grain size distribution, created a hand-drawn map of the surface patch, and calculated the boundary shear stress. To validate our numerical model, we use only the high-resolution riverbed elevation, automated surface grain size distribution, and hand-drawn map of the surface patch.

We set the same channel geometry and sediment grain size distribution as those used in the experiment. The channel was discretised into 110 cells in the longitudinal direction and 25 cells in the transverse direction. Because the reference study did not mention the sediment density, we assumed a density of 2650 kg/m³. The porosity of the bed was 0.4. The computational time was 20 h, and the flow discharge was fixed at 0.4 m³/s to achieve an equilibrium state of the bed and texture. We also constricted the flow at the upstream end by setting one-third of the cells on the right bank side as obstacle cells to mimic the concrete block placed at the upstream in the experiment. To reproduce the sediment recirculation, the amount and distribution of sediment runoff from the downstream end were given equally to the cells at the upstream end, except for the obstacle cells, in the next time step. In the experiment, there may have been a time lag in conveying the sediment from the downstream end to the upstream end; however, we did not consider this time lag.

Fig. 2 shows the change in water depth and bed elevation from the initial bed to the end of the simulation using the bed geometry observed in the experiment by Nelson et al. (2010). Two large bars are observed: the upstream bar is on the left side of the channel between 20 and 35 m from the upstream end, and the downstream bar is on the right side of the channel between





40 and 55 m from the upstream end. Both bars partially emerge above the water surface (with a depth of less than 0.02 m). Deep pools form on the opposite banks of both bars. The numerical results can generally replicate the bar shape and wavelengths, although the model appears to overpredict the bar height.

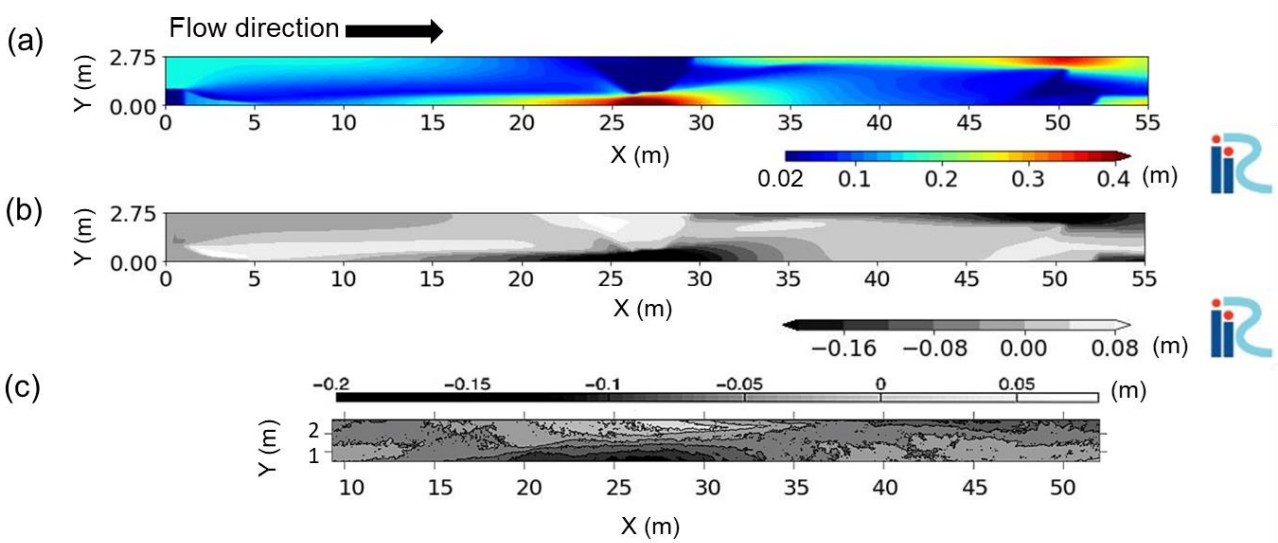

Fig. 2 (a) Two-dimensional bathymetry from our calculation at the end of the calculation. (b) Two-dimensional riverbed profiles from our calculation at the end of the calculation. (c) Two-dimensional riverbed profiles from the experiment by Nelson et al. (2010), which is adapted from Nelson et al. (2010).

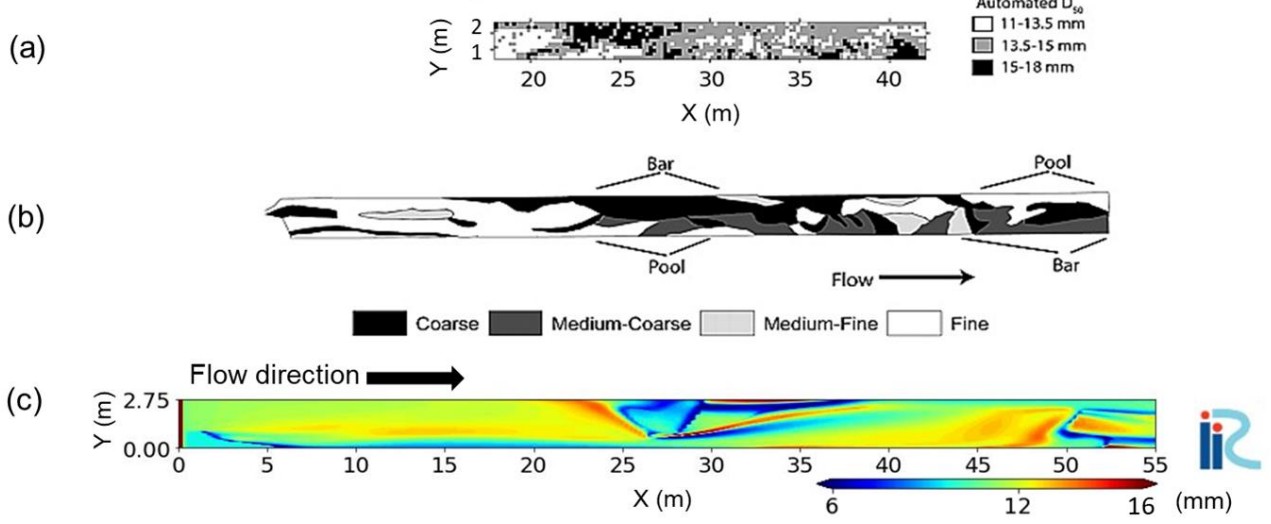

Fig. 3 (a) The automated map of the local grain size $d_{50}$ for which 50% of the grain size distribution is finer from the experiment by Nelson et al. (2010). (b) Hand-drawn surface patch map from the experiment by Nelson et al. (2010). (b) and (c) are adapted from Nelson et al. (2010). (c) The map of $d_{50}$ from our calculation at the end of the calculation.





Fig. 3 (a) and (b) show the distribution of the surface median grain size and the hand-drawn surface patch map from the experiment, respectively. In this experiment, coarse bars and fine pools developed. Several studies have suggested that this surface sorting pattern is typical for alternate bars developed in a straight channel [e.g. Lisle and Hilton, 1999; Recking et al., 2016]. Nelson et al. (2010) concluded that this is because "along a path moving up the bar, the material moving as cross-stream
sediment transport became finer, preferentially shuttling fine sediment into the pools". Fig. 3 (c) shows the map of the surface median grain size based on the numerical results. The computational results are generally consistent with the experimental results, i.e. showing coarse-grained bars and fine-grained pools. One discrepancy between the simulation and experiment is the formation of an extremely fine-grained, emerged bar. This may be because the emerged bar is calculated to have zero sediment transport capacity, and thus fine particles that would normally flow down to pools are instead deposited there. This
is a limitation of the shallow-water equation and equilibrium sediment transport model used in this study. Apart from this feature, the numerical model has sufficient accuracy for simulating the grain size characteristics over the alternate bars observed in the experiment.

## 3 Results

### 3.1 Calculation conditions

Herein, we investigate the effect of grain sorting waves caused by non-equilibrium sediment supply on the free-migrating alternate bars in the poorly sorted sediment bed using the iRIC-Nays2DH morphodynamic model, as validated above. To clearly show the presence of grain-sorting waves and quantify their effect on the bar dynamics, we follow the HBL concept proposed by Wong and Parker (2006) and its breakdown in the poorly sorted sediment case noted by An et al. (2017) in the numerical experiments. In other words, the unsteady, symmetrical triangular water discharge hydrograph and constant
sediment supply given in the upstream boundary under poorly sorted sediment generate a low-amplitude, grain-sorting wave that migrates downstream beyond the typical length scale of the HBL recognised in well-sorted sediment beds. As an example of poorly sorted gravel-bed sediment rivers, we consider the conditions of the Otofuke River, as in Dai et al. (2021) and Huang et al. (2023), which provides maximum and minimum discharges of 1200 and 100 m$^3$/s, respectively, with a duration, $T_h$, of 80 h. The channel geometry is 21 km in length, 70 m in width, and has a slope of 0.00541.
Four calculations are performed under this general computational setting (Table 1), focusing on the sediment size distribution range and presence of alternate bars. We determine the sediment size distribution based on field data obtained from the Otofuke River in 2016 [Kyuka et al., 2020]. Fig. 4 shows a wide sediment size distribution range of 0.4 mm to 200 mm, which is typical of poorly sorted sediment in gravel-bed rivers (Fig. 4). We define this case as the base case (i.e. Case 1); to understand the effect of the size distribution range, we perform an additional morphodynamic calculation that uses poorly
sorted sediment but a narrower grain size range than that of Case 1. For this purpose, we employ the method proposed by An et al. (2017). First, we prepare the original data for sediment distribution and specify grain sizes in the $\psi$ logarithmic scale as follows:





Table. 1 Summary for calculation case.

| Case | Channel geometry | Width (m) | Sediment data | Discharge (m³/s) (max,min) | Sediment supply (m²/s) |
|------|------------------|-----------|---------------|----------------------------|------------------------|
| 1-b | Bar | 70 | $\xi = 1$ | 1200, 100 | 0.00335 |
| 1-n | Non-bar | 7 | $\xi = 1$ | 120, 10 | 0.00335 |
| 2-b | Bar | 70 | $\xi = 0.5$ | 1200, 100 | 0.0027 |
| 2-n | Non-bar | 7 | $\xi = 0.5$ | 120, 10 | 0.0027 |

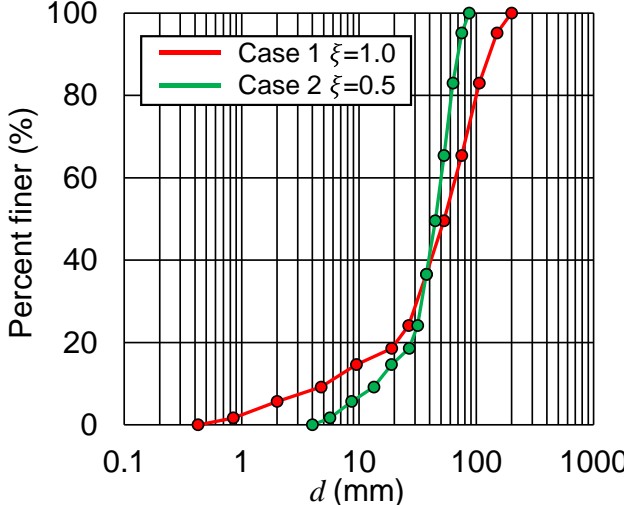

Fig. 4 Grain size distribution of $\xi = 1.0$ (Cases 1-b and 1-n) and $\xi = 0.5$ (Cases 2-b and 2-n).

$$\psi_i = \frac{\ln d_i}{\ln 2}. \qquad (19)$$

The original grain size distribution is specified as the pairs of $(\psi_i, F_i)$, where $F_i$ is the fraction by weight of sediment finer than size $\psi_i$. We can specify the group of grain size distributions as the pairs of $((\psi_i - \psi_m)\xi + \psi_m, F_i)$, where $\psi_m$ is the arithmetic mean grain size on $\psi$, and $\xi$ is a user-specified coefficient. We can vary the range of sediment distribution by changing $\xi$; its value is set to 0.5 (Case 2) in this study (Fig. 4). The original size distribution corresponds to $\xi = 1$. Both distributions have the same geometric mean grain size, $d_g$ (=37.66 mm), but they have different standard deviations, $\sigma_g$ ($\xi = 1: 3.60, \xi = 0.5: 1.90$); importantly, both are classified as poorly sorted sediments. Note that the case with $\xi > 1$, which is a quite poorly sorted sediment bed, is not tested here because this condition causes the presence of quite large sediments, which are not movable in the hydrograph condition defined in this study, resulting in significantly different bar migration features.

In addition to the above cases, we also perform corresponding one-dimensional calculations to demonstrate the fundamental features of grain-sorting wave migration without alternate bars. To simulate this, we use a narrower channel but the same unit discharge employed in the two-dimensional calculation to restrict the bar regime to a flat bed. Note that for this



narrower channel case, we still use the two-dimensional morphodynamic model, iRIC-Nays2DH, for consistency with the alternate bar cases. The calculation conditions of these runs in terms of the grain size distribution and presence of alternate bars are summarised in Table 1. For both the base and narrow channel cases, we use identical grid sizes in both spatial directions: the base channel (bar case: Case O-b) and narrow channel (non-bar case: Case O-n) are discretised into $600 \times 20$ cells and $200 \times 2$ cells, respectively. The porosity of the bed is 0.4.

The constant sediment supply rate in the simulation is determined by a combination of the channel slope, hydrograph, and sediment size distribution. In the simulations, we fix the hydrograph, channel slope, and sediment size distribution and adjust the sediment supply rate from the upstream end to achieve macro-scale morphodynamic equilibrium, i.e. the only variation during the hydrograph is upstream bed fluctuation and migration of the grain-sorting wave, while the macroscale bed slope is maintained. This restriction produces sediment transport rates from the upstream end of 0.0027 and 0.00335 $m^2$/s for

Cases 1 and 2, respectively. The grain size distribution of the supplied sediment is the same as that of the initial riverbed.

## 3.2 Calculation results

We address the results of the non-bar cases (i.e. Cases 1-n and 2-n) first to show the fundamental characteristics of the formation and migration of the grain-sorting wave. Fig. 5 shows the detrended riverbed elevation (difference from the exact equilibrium riverbed slope) and the geometric mean grain size along the right bank within the last single hydrograph for which

the macroscopic equilibrium state was satisfied. Note that there is a riverbed change near the downstream end owing to the downstream end conditions (i.e. the uniform flow assumption). For the equilibrium riverbed slope, we employ the average riverbed slope in the range of 3,000–18,000 m at the end of the calculation (4000 h), excluding the river reaches close to the upstream and downstream ends, which have large-scale riverbed fluctuations. The results of Case 1-n show that the large bed elevation change caused by the non-equilibrium sediment supply is limited to within 1 km from the upstream end, similar to

the HBL observed in the well-sorted sediment case (Wong and Parker, 2006). In addition to the large-scale riverbed fluctuation within this limited reach, a sediment wave of grain size order migrates downstream through the entire channel with diffusion. An et al. (2017) suggested that this sediment wave is a grain-sorting wave "bedload sheet", which is formed by the imbalance between the sediment supply and sediment transport capacity. Fig. 5 shows that the geometric mean grain size is relatively small at the centre of the bedload sheet. In other words, the effects of the non-equilibrium sediment supply at the upstream end

are conveyed over long distances downstream through the migration of bedload sheets, indicating a breakdown of the HBL concept in rivers with poorly sorted sediment riverbeds [An et al., 2017]. Fig. 5 (b) shows that the HBL-like upstream river reach in Case 2-n is longer than that in Case 1-n because of the larger sediment supply. This is consistent with the results of the theoretical analysis by Wong and Parker (2006). In contrast, in Case 2-n, the bedload sheet can migrate a long distance downstream, as in Case 1-n, but the presence of the bedload sheet is somewhat unclear, i.e. the amplitude of this wave and the

associated grain size difference are much smaller than that of Case 1-n because of the narrow range of the sediment size distribution (Fig. 4). This implies that the grain-sorting wave in Case 1 may have a larger impact on the downstream morphodynamics than that in Case 2. We will investigate this in the two-dimensional calculations with alternate bars below.





Fig. 5 The detrended riverbed elevation (difference from the exact equilibrium riverbed slope) and the geometric mean grain size, $d_g$, along the right bank within the last single hydrograph: (a) Case 1-n; (b) Case 2-n. The yellow area indicates the HBL-like reach.

We then show how this feature differs in the two-dimensional cases under the presence of migrating alternate bars; in other words, we investigate how long-migrating grain-sorting waves impact the downstream alternate bar dynamics. Fig. 6 shows that in both cases, alternate bars are formed from x = 3 km and migrate downstream. Fig. 7 shows the longitudinal riverbed variation from the initial riverbed elevation and the geometric mean grain size along the right bank within the last single hydrograph under the macroscopic equilibrium state. It is clear that the bedload sheet migrates downstream, as in the non-bar cases, but the behaviour of the bedload sheet within alternate bars is unclear because the structure of the bars is approximately two orders of magnitude larger than that of the bedload sheets.

Figs. 8 and 9 display the planimetric riverbed variation and geometric mean grain size within the last single hydrograph (more specifically, t = $0T_h$, $0.25T_h$, $0.5T_h$, and $0.75T_h$) in the upstream (3–5 km) and middle reaches (10–12 km). Regardless of the time, there are coarse bars and fine pools, which are typical surface textures for alternate bars in straight channels [e.g.



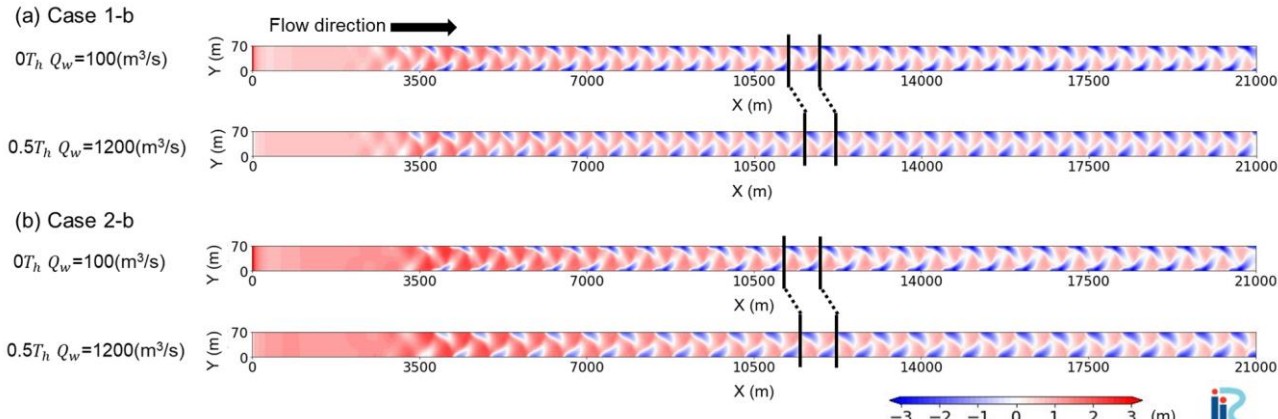

Fig. 6 Two-dimensional riverbed variation from initial riverbed elevation at $0T_h$ (upper row) and $0.5T_h$ (lower row): (a) case 1-b; (b) case 2-b.

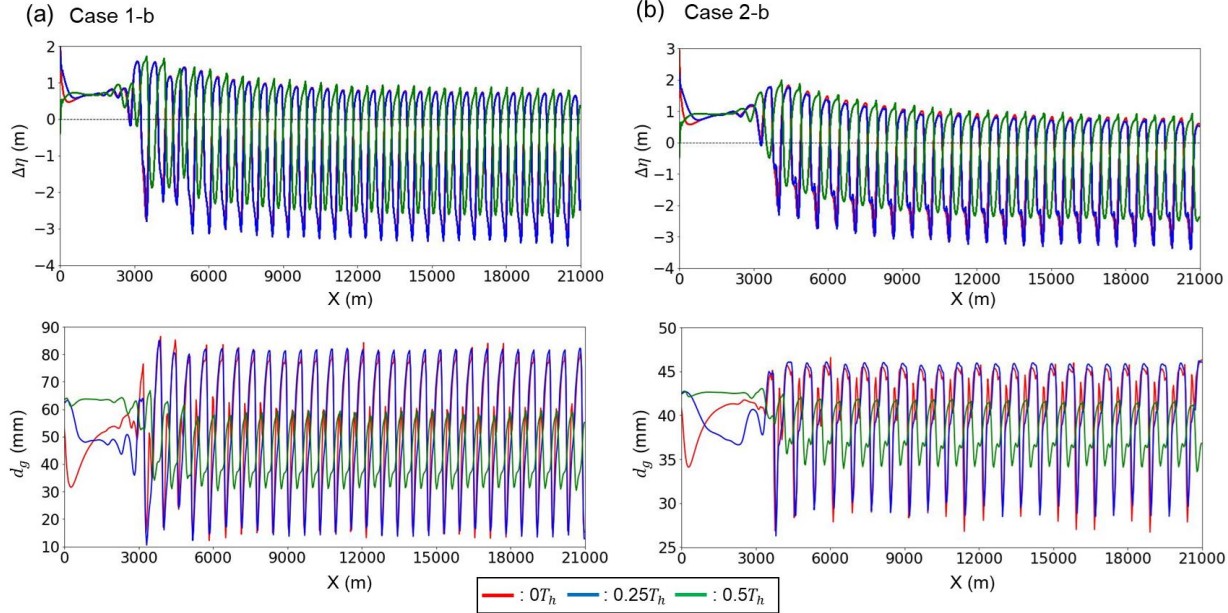

Fig. 7 The longitudinal riverbed variation from the initial riverbed elevation, $\Delta\eta$, and geometric mean grain size, $d_g$, along the right bank within the last single hydrograph: (a) Case 1-b; (b) Case 2-b. Each colour (red, blue, green) indicates a distribution at each time ($0T_h$, $0.25T_h$, $0.5T_h$). Note that bedload sheets cannot be visualized in the upper figure ($\Delta\eta$) because this figure is focused on bar configuration.

Lisle and Hilton, 1999; Nelson et al., 2010; Recking et al., 2016]. Coarse patches are formed at the minimum flow discharge ($0T_h$), and then these patches are flushed as the flow discharge increases [Hassan and Church, 2001; Mao, 2012]; thus, the

maximum flow stage ($0.5T_h$) has the smallest geometric mean grain size in a single hydrograph. Comparisons of the two

 

Fig. 8 Two dimensional distributions of riverbed change (upper low) and surface geometric mean diameter (lower low) at each time (($0T_h$, $0.25T_h$, $0.5T_h$, $0.75T_h$) in Case 1-b.

reaches (i.e. the upstream and middle reaches) illustrate that the middle reach has regular bar shapes, whereas the upstream reach has slightly irregular shapes. A more evident difference in the morphodynamic features between the upstream and downstream reaches is the surface texture of the rising limb ($0.25T_h$) and falling limb ($0.75T_h$). In general, the surface texture becomes coarser at the rising limb owing to coarse patches formed at the minimum flow discharge [e.g. Mao, 2012], which is

seen in the middle reach where the bar shape is regular. However, the upstream reach exhibits a finer surface texture at the rising limb because the migrating bedload sheet reaches the upstream bars, causing a large supply of fine particles. To confirm this bar shape difference more quantitatively, we conduct wavelet analysis to detect the spatial change in the dominant bar length. Wavelet analysis was introduced by Grossmann and Morlet (1984) to treat geophysical seismic signals, and it can







Fig. 9 Two dimensional distributions of riverbed change (upper low) and surface geometric mean diameter (lower low) at each time (($0T_h$, $0.25T_h$, $0.5T_h$, $0.75T_h$) in Case 2-b.

accurately analyse unstable signals. Only a few studies have employed this method with respect to river morphology, but

Huang et al. (2023) used wavelet analysis to investigate the local migration period in alternate bars, and this method is fully applicable to the calculation of the wavelength in alternate bars. Fig. 10 shows the results of the wavelet analysis of the dominant wavelength along the right bank at $0T_h$ and $0.5T_h$ in the last single hydrograph. The results show a strong peak in the middle and downstream reaches, such that the dominant bar length is consistent in space in this reach. The wavelength of Case 1-b, which has more poorly sorted sediment, is approximately 600–650 m, which is shorter than that of Case 2-b

(approximately 750 m). This relationship between the sediment features and wavelength agrees with the linear stability analysis performed by Lanzoni and Tubino (1999). However, in the upstream reach, although a strong peak occurs, we can also





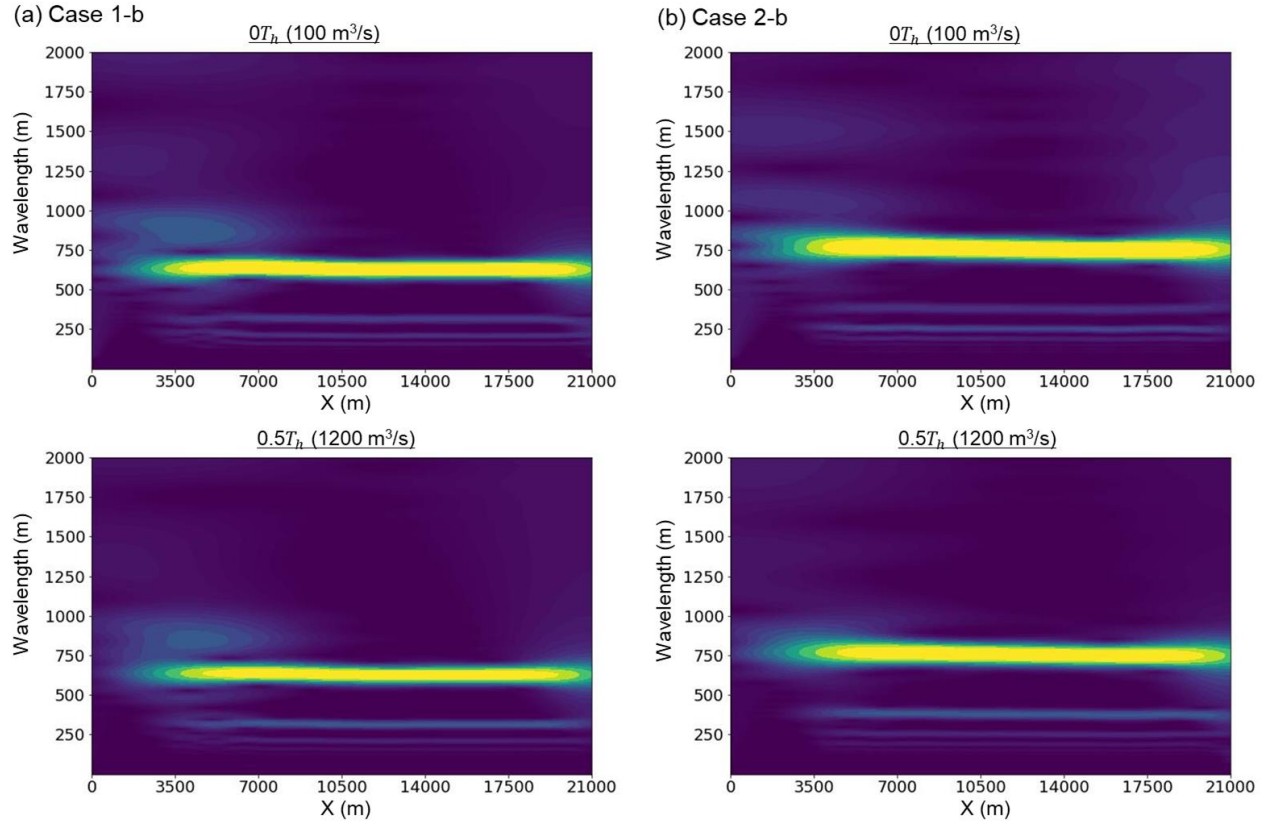

Fig. 10 Wavelet analysis of the dominant wavelength along the right bank at $0T_h$ and $0.5T_h$ within the last single hydrograph: (a) Case 1-b, (b) Case 2-b.

recognise secondary peaks around the dominant peak, indicating that the bar shape is more irregular than that of the middle reach. Importantly, this indication of an irregular bar is not evident in Case 2-b, which a relatively better sorted sediment than in Case 1-b. This indicates that a grain-sorting wave with some degree of finer/coarser features may impact the alternate bar
dynamics.

To quantify the behaviour of the bedload sheets within the bars, we examine the longitudinal distribution of the sediment flux for each grain size. Figs. 11 and 12 show the longitudinal distributions of the cross-sectional average bedload transport flux, $\overline{q_{B_t}}^{xy} (= \overline{\sqrt{(q_{B_t}^x)^2 + (q_{B_t}^y)^2}})$, for each grain size in Cases 1-b and 2-b, respectively. Fig. 11 shows a strong temporal variation in the sediment transport rate corresponding to riverbed change in the upstream reach, which is also observed for
well-sorted sediments [e.g. Wong and Parker, 2006]. In addition, the local peak of $\overline{q_{B_t}}^{xy}$ migrates downstream as a bedload sheet in the early stage of the rising limb of the hydrograph ($0.1T_h$–$0.2T_h$), and then reaches the train of alternate bars. However, downstream of 7 km, these small variations are absent, and the sediment flux remains constant in space, indicating that this reach is in a dynamic equilibrium state. Note that the small fluctuations seen at x = 3.5–21 km are due to bars. This indicates




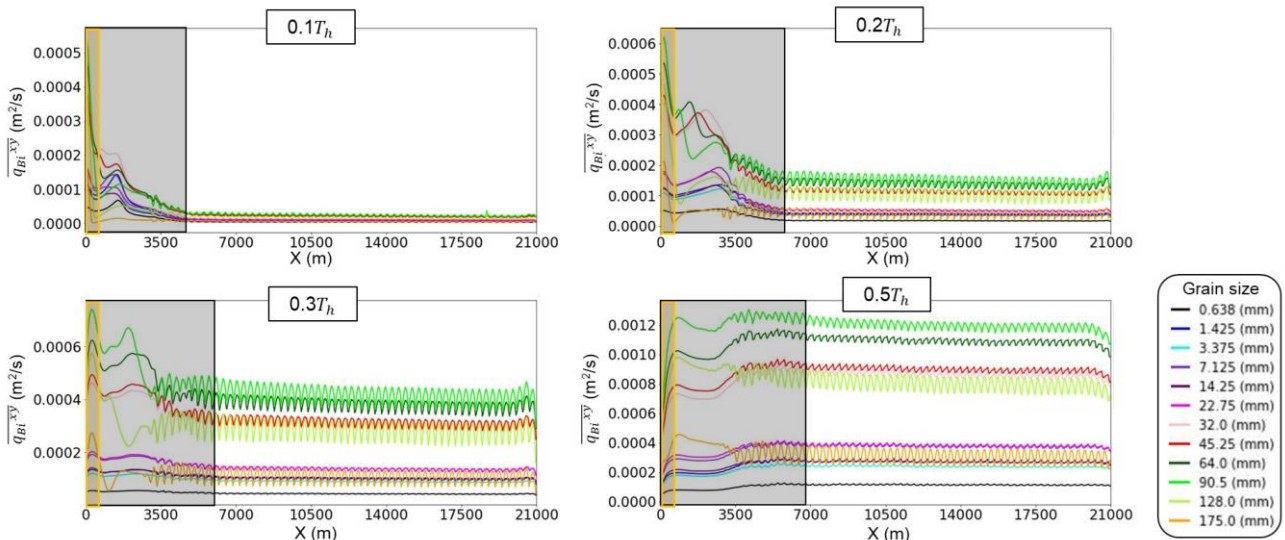

Fig. 11 The longitudinal distribution of the magnitude of cross-sectional average sediment transport flux, $\overline{q_{Bi}}^{xy}$ (= $\sqrt{(q_{Bi}{}^x)^2 + (q_{Bi}{}^y)^2}$), for each grain size at each time ($0.1T_h$, $0.2T_h$, $0.3T_h$, $0.5T_h$) in Case 1-b. The yellow area and the grey area indicate the HBL-like reach and the affected length by bedload sheets, respectively.

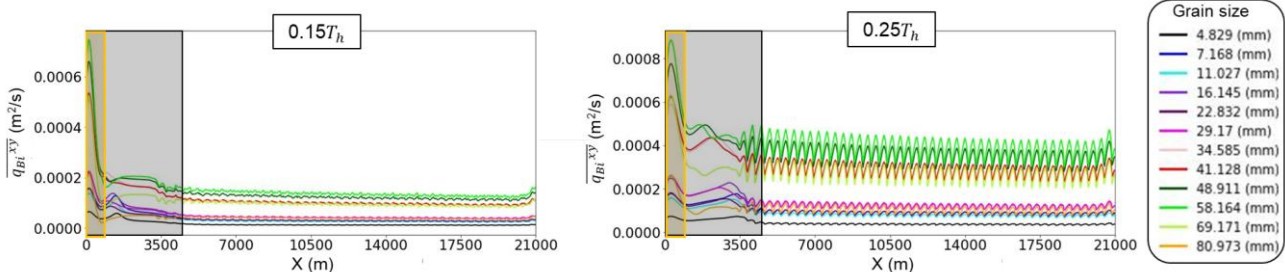

Fig. 12 The longitudinal distribution of the magnitude of cross-sectional average sediment transport flux, $\overline{q_{Bi}}^{xy}$ (= $\sqrt{(q_{Bi}{}^x)^2 + (q_{Bi}{}^y)^2}$), for each grain size at each time ($0.15T_h$, $0.25T_h$) in Case 2-b. The yellow area and the grey area indicate the HBL-like reach and the affected length by bedload sheets, respectively.

that the bedload sheets affect the sediment transport rate until x = 7 km in the rising limb of the discharge, after which they eventually dissipate in the entire reach at $0.5T_h$. Because this length is consistent with the reach that shows bar irregularity, as shown in Fig. 10 (a), it may suggest that bedload sheets can impact bar characteristics, including the wavelength, in this reach. However, unlike the non-bar case, the bedload sheet disappears as it migrates within the bar area because the bar structure is larger than that of the bedload sheets. Although Fig. 11 shows the dissipation of bedload sheets at $0.5T_h$, an irregular bar shape still exists (Fig. 10 (a)). This suggests that the impact of bedload sheets on the bar shape, such as the wavelength, lasts longer

than the lifetime of the bedload sheets themselves. In Case 2-b, the length affected by the bedload sheets also extends to approximately 4 km, where the surface texture is irregular (Fig. 12), meaning that the affected length in Case 2-b is shorter





than that in Case 1-b (Fig. 10 (b)). This may be because the structure of the bedload sheets in Case 2-b has a narrow grain size distribution range, and the associated effect on the bar dynamics is smaller than that in Case 1-b.

Previous studies have suggested that bedload sheets disturb sediment transport [Whiting et al., 1988; Venditti et al., 2008;
Nelson et al., 2009; Recking et al., 2009]. Fig. 13 shows the temporal variation in the flow discharge and the cross-sectional average sediment transport flux, $\overline{q_B}^{xy}$ ($= \sum \overline{q_{Bi}}^{xy}$), in the last single hydrograph under the equilibrium state. In Case 1-n, where the bedload sheets are evident without bars, the bedload sheet migration increases the sediment transport rate, causing hysteresis between the water discharge and sediment transport rate. On the other hand, Case 1-b exhibits a weak counterclockwise (CCW: the flow peak leads the sediment transport peak) hysteresis because of the disappearance of bedload
sheets in the bar reach, where a spatially constant bedload transport rate is achieved (i.e. downstream of 6300 m in Fig. 11). Case 2-n, which has small grain size distribution range ($\xi = 0.5$), also exhibits a small disturbance induced by bedload sheets at 2100 m (Fig. 13). The contribution of bedload sheets is negligible when comparing Cases 2-n and 2-b at 11,340 m and 16,170 m, even though the bedload sheets migrate to the downstream end in Case 2-n (Fig. 5). These results are attributed to the smaller magnitude of the bedload sheets in Case 2-n compared with those in Case 1-n, suggesting that the magnitude of
the bedload sheets also contributes to the affected length because of the non-equilibrium sediment supply from the upstream end.

## 4 Discussion

The focus of this study is to clearly understand the effect of sediment supply conditions in poorly sorted sediment on downstream river morphodynamics and the corresponding grain size distribution. Thus, we employ the concept of the HBL as
an effective spatial scale for the non-equilibrium sediment supply from the upstream end. Although this study uses simplified upstream conditions (a symmetric triangular-shaped hydrograph and constant sediment supply) to create the HBL, this computational setting can partly represent morphodynamic features that may occur under conditions of an unsteady flow and non-equilibrium sediment supply.

Under upstream conditions of symmetric triangular-shaped hydrographs and a constant sediment supply, bedload sheets,
which are a type of grain-sorting wave, are formed within the HBL and migrate far downstream from the upstream end (Figs. 5 and 7). These bedload sheets are not due to instability of the riverbed [Seminara et al., 1996] but are formed because of an imbalance between the sediment supply and sediment transport capacity [An et al., 2017]. This is consistent with the characteristics of bedload sheets, which have grain-scale coarse tips and a zone behind filled with fine particles within the coarse particles, as observed in the field [Whiting et al., 1988] and in experiments [Kuhnle and Southard, 1988; Venditti et al.,
2008; Nelson et al., 2009; Recking et al., 2009]. The bedload sheets simulated in our numerical experiments are also this type of morphodynamic feature. Furthermore, the characteristics of bedload sheets depend on the sediment transport and distribution of the riverbed [An et al., 2017], and their magnitude contributes to the effect on the downstream bar morphology (Figs. 10, 11, and 12). However, this study is applicable only to gravel-bed rivers with poorly sorted sediment; thus, different





Fig. 13 The temporal variation of flow discharge and the magnitude of cross-sectional average sediment transport flux, $\overline{q_B{}^{xy}}$, in last single hydrograph: (a) Case 1-n; (b) Case 1-b; (c) Case 2-n; (d) Case 2-b. The red line and the blue line indicate flow discharge, $Q_w$, and sediment transport flux, $\overline{q_B{}^{xy}}$, at rising limb and falling limb, respectively.

phenomena will occur in rivers with well-sorted sediment or those dominated by suspended sediment.

395        The migration of bedload sheets changes the mobility of the sediment, which affects only the alternate bar morphology





located upstream; however, the bedload sheets disappear as they migrate through the bar reach (Figs. 11, 12, and 13). This indicates that the river reach length affected by the bedload sheet is limited, and bedload sheet migration has little effect on most parts of the alternate bars in our simulation. Several studies have reported similar morphodynamic characteristics. For instance, Lisle et al. (1997) reported that sediment pulses had little effect on the dynamics of alternate bars. It should be noted

that they used well-sorted sediments; however, our results agree with their findings. Nelson et al. (2015) concluded that a riffle-pool structure played a role in dissipating sediment pulses. Although the riffle-pool and alternate bars are different bedforms, their experimental results support our results in that the three-dimensional bedform structure disperses migrating sediment waves caused by non-equilibrium sediment supply conditions. Iwasaki et al. (2017), who numerically clarified the dynamics of bedload particle tracers in alternate bars, claimed that migrating alternate bars significantly affected the tracer

movement, resulting in superdiffusion of the tracer, which led to much faster sediment dispersal than normal dispersion. These studies and the current numerical results show that sediment mixing and dispersal due to migrating alternate bars are the main causes of bedload sheet dissipation within short distances and the inhibition of further downstream migration. On the other hand, Humphries et al. (2012) experimentally observed the sediment pulse dynamics on fixed alternate bars that were immobilised using sandbags to prevent exposure to sediment pulses. Their results indicated that sediment pulses mainly

migrated to the channel pool characterised by the fixed alternate bars as if bypassing the fixed bars. Although the pulse celerity varied locally owing to the local flow features forced by the alternate bars, the sediment pulse could migrate further downstream. The morphological features of large-scale bedforms, such as alternate bars and their dynamics (i.e. mobile or immobile), play a critical role in the migration of bedload sheets.

In contrast to our study, in which the impact of the non-equilibrium sediment supply on bar dynamics was limited, many

experimental studies have argued that there are strong impacts from the sediment inflow [Podolak and Wilcock, 2013; Bankert and Nelson, 2018; Nelson and Morgan, 2018] or cutoff [Lisle et al., 1993; Venditti et al., 2012]. A much larger and longer effect of sediment supply/reduction will eventually change the alternate bar dynamics. Although the effects of sediment supply are likely to be propagated owing to the limited flume length, the critical difference between our study and these previous studies is the time scale. Many parts of previous studies have focused on the impact of permanent changes in sediment supply

conditions; however, our study targets the impact of the ephemeral non-equilibrium sediment supply in a single hydrograph. Long-scale changes are beyond our scope, but our results may be useful for distinguishing short- and longer-scale effects of sediment sources on river morphodynamics.

The triangular hydrograph and passage of bedload sheets cause hysteresis in the sediment transport (Fig. 13). Weak counterclockwise (CCW: the flow peak leads the sediment transport peak) hysteresis is observed in reaches where no bedload

sheets exist. Gunsolus and Binns (2017), who reviewed sediment transport hysteresis, mentioned that CCW hysteresis is common in rivers where bedload transport is dominant [e.g. Bombar et al., 2011]. Several studies have asserted that the development of bedforms such as dunes causes CCW hysteresis [Lee et al., 2004; De Costa and Coleman, 2013; Martin and Jerolmack, 2013; Waters and Curran, 2015]. However, our shallow water flow model cannot represent dune morphology; therefore, bedform development is not the factor driving our CCW hysteresis. Wang et al. (2019) reported that short-term



hydrographs such as flash floods cause CCW hysteresis. This is because the sediment transport regime is unable to respond to changes in the flow regime in a short duration, i.e. the hysteresis observed in our computational results may also be due to a hydrograph with a short duration. In contrast, bedload sheets migrate downstream only during the rising limb of the hydrograph, leading to a strong ephemeral clockwise (CW: the sediment transport peak leads the flow peak) hysteresis. Humphries et al. (2012) reported that CCW hysteresis was observed with sediment pulses because of the lag caused by the transport distance

between the source and measurement points. However, after sediment pulse injection, there was a large amount of available sediment in the channel, resulting in CW hysteresis. Our hysteresis due to the ephemeral increase in sediment transport induced by bedload sheets supports their findings, suggesting an indirect effect of the sediment pulse on hysteresis. Furthermore, the combination of CCW hysteresis and ephemeral CW hysteresis results in figure-eight hysteresis. A few cases of figure-eight hysteresis have been reported [Williams, 1989; Waters and Curran, 2015], but no clear reasons for these phenomena have been

noted. Our numerical results suggest that the grain-sorting wave itself contributes to sediment transport hysteresis, and figure-eight hysteresis occurs during the passage of grain-sorting waves; however, the presence of alternate bars suppresses this hysteresis. This indicates that not only the flow regime but also the interaction among different morphological features, such as grain-sorting waves and alternate bars, play key roles in the sediment transport characteristics, such as hysteresis.

The computational results indicate that the migration of bedload sheets generated by a single flood hydrograph event has

a limited effect on the alternate bar dynamics. This is valid for this spatiotemporal scale but is surely dependent on the flow regime, intensity of the sediment source impact, and sediment composition of the riverbed and feeding. For instance, the amount of sediment supply affects the size of the HBL [Wong and Parker, 2006] and the migration celerity of bedload sheets [Nelson et al., 2009]. Venditti et al. (2008) reported that bedload sheets are formed only when the sediment supply is reasonably close to the sediment transport capacity and all particles are in a fully mobile state. As the shear stress on the riverbed increases,

bedload sheets either transition into dunes [e.g. Whiting et al., 1988] or disappear [Recking et al., 2009]. In addition, the compositions of the riverbed and sediment supply also significantly contribute to determining the sediment mobility [e.g. Wilcock and Crowe, 2003] and bar characteristics [e.g. Lanzoni and Tubino, 1999]. Fine sediment improves the mobility of coarse sediments because the fine sediment fills the interstices between coarse sediments and reduces the resistance of the riverbed surface, which is called the magic sand effect [e.g. Wilcock, 1998; Wilcock et al., 2001]. In this case, bedload sheets

deliver more fine-grained sediment, contributing not only to bar shapes, but also to bar mobility [Podolak and Wilcock, 2013; Bankert and Nelson, 2018]. Because bedload sheets and fluvial bars are sensitive to external forces, different hydrographs and sediment supplies may cause different morphodynamic phenomena [e.g. Gaeuman, 2014; Peirce et al., 2019]. Finally, the dynamics of large-scale morphological features such as alternate bars also affect the dispersal of bedload sheets. This study addresses only migrating alternate bars, but Iwasaki et al. (2017) indicated that the dispersal patterns of the incoming sediment

from upstream differ between migrating and non-migrating bars. Fixed bars are more likely to store the incoming sediment, meaning that migrating and non-migrating bars may interact differently with bedload sheets. Furthermore, in the presence of other bed morphologies (e.g. multiple-row bars and braiding), the bedload sheet dynamics and interactions with the respective bedforms will differ from those of alternate bars. These complexities related to the hydrograph, sediment supply, texture, and



morphological features may play key roles in controlling the morphodynamic features targeted in the current study, suggesting
the need for further studies to understand large-parameter spaces in the future.

**5 Conclusion**

In this study, we present numerical simulations of the interaction between alternate bar dynamics and the migration of bedload
sheets in poorly sorted sediment to understand the morphological response of alternate bars to non-equilibrium sediment supply
conditions. More specifically, we perform two-dimensional morphodynamic calculations using iRIC-Nays2DH in a straight
channel under a repeated cycle of an unsteady water hydrograph and a constant supply of poorly sorted sediment. In the well-
sorted sediment cases, the upstream non-equilibrium sediment supply can propagate only a limited distance from the upstream
end [i.e. the hydrograph boundary layer, Wong and Parker, 2006]. However, a poorly sorted sediment breaks down the HBL
concept, meaning that low-amplitude bedload sheets generated by non-equilibrium sediment supply conditions propagate far
downstream [An et al., 2017]. In this context, the upstream water and sediment boundary conditions may affect the far-
downstream river dynamics through the migration of bedload sheets. The aim of this study is to quantify the effect of this type
of bedload sheet on the downstream river morphology, specifically on alternate bars. This does not mimic the specific situation
in natural streams; rather, we aim to represent the morphodynamic response of gravel-bed rivers with poorly sorted sediment
to the upstream forcing condition in which the sediment supply volume and sediment transport capacity do not match under
unsteady flow conditions.
480        The numerical results show that clear bedload sheets migrate downstream in the poorly sorted sediment case and impact
the train of alternate bars that develop in the downstream reach. More specifically, the bedload sheets supply fine sediment to
the alternate bars, contributing to a change in the surface texture of the bars and irregularity of the bar characteristics (i.e. the
wavelength). This change in bar characteristics is unclear in the case of a narrower grain size distribution range, which causes
the migration of bedload sheets; however, its intensity is much weaker. This suggests an important effect of bedload sheets on
the downstream alternate bars, and further suggests that the upstream non-equilibrium sediment supply condition has a non-
negligible role in downstream river morphodynamics even far from the sediment feed point. However, this effect of the bedload
sheets on the bars does not propagate across the entire channel and disappears completely in the alternate bars located further
downstream. The alternate bars of such a downstream reach show regular patterns in terms of their shape factor, indicating a
limited or negligible effect of bedload sheets. This is because the structure of the bars is approximately two orders of magnitude
larger than that of the bedload sheets; therefore, the bedload sheets are strongly dispersed by the migration of the alternate bars.
This suggests that the bedload sheets generated by an imbalance between the upstream sediment supply and transport capacity
have a limited effect on the downstream river morphodynamics, as long as active and larger morphological changes, such as
alternate bars, are the dominant morphodynamic features in the targeted river reach.
        Our study was performed under a limited combination of parameters, such as the hydrograph, sediment supply conditions,
and grain size distribution; therefore, a wider range of parameters should be further tested to confirm our results. In addition,



although our findings should be interpreted as a short-term scale effect of upstream boundary conditions on the downstream river morphology, a much longer-term, persistent effect of upstream boundary conditions will be dominant from a long-term perspective. Nevertheless, our results can provide useful insights into the combination of such short- and long-term effects of the upstream water–sediment condition on the downstream river system.

## Data availability

The data used to support the findings of this study are available from the corresponding author upon request.

## Author contributions

Soichi Tanabe - Conceptualization; funding acquisition; methodology; investigation; software; writing—initial draft.
Toshiki Iwasaki - Conceptualization; funding acquisition; methodology; resources; software; supervision; writing—initial draft; writing—reviewing and editing.

## Competing interests

The authors declare no competing interests.

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
