# Peer review of "Effect of grain-sorting waves on alternate bar dynamics: Implications of the breakdown of the hydrograph boundary layer"

_EGUsphere, 2025_

## Referee Comment (RC1)

**Reviewed Manuscript:** Effect of grain-sorting waves on alternate bar dynamics: Implications of the breakdown of the hydrograph boundary layer

**Authors:** Soichi Tanabe, Toshiki Iwasaki

**Journal:** Earth Surface Dynamics

**Referee:** Chenge An (anchenge08@163.com)

It has been known that with cycled hydrographs, the alluvial river can reach a dynamic equilibrium under which only a limited distance from the inlet is influenced by the varying inflow discharge. This region is called a hydrograph boundary layer (HBL), and can be break down by advective bedload sheets when sediment is poorly sorted. However, previous studies only apply a 1D analysis, and cannot consider the alternate bars that are commonly observed in natural river. In this study, the authors extend previous studies to a 2D framework, and can therefore study the interaction of bedload sheets and alternate bars with the breakdown of HBL. That said, I regard this manuscript as a big progress on the knowledge of the HBL as well as the dynamic equilibrium of alluvial rivers. The topic is suitable for Esurf. Generally, the manuscript is well organized and easy to read. However, I still find a few issues that need to be addressed, with the major issue being about the governing equation. I list my detailed comments below.

**Main comments:**

1. Equation (7) seems to be incorrect to me. Following is the equation from Chapter 4 of Parker's e-book (Parker, 2004). A major difference between your Eq. (7) and the following equation from Parker's e-book is that, Eq. (7) does not have $f_{Ii}$ which represent the grain size distribution of the sediment exchanging between the active layer and the substrate material. Besides, I am also confused with the L167-168 of Page 6. Doesn't $F_{ai}$ denote the grain size distribution of the active layer by definition?

$$(1-\lambda_p)\left\{ f_{Ii}\left( \frac{\partial}{\partial t}(\eta - L_a) + \sigma \right) + \frac{\partial}{\partial t}(F_i L_a) \right\} = -\vec{\nabla} \cdot \vec{q}_{ti}$$

Parker, G. (2004). *1D sediment transport morphodynamics with applications to rivers and turbidity currents.* Retrieved from http://hydrolab.illinois.edu/people/parkerg//morphodynamics_e-book.htm

**Specific comments:**

1. L20-24 P1: In the last two sentences of the abstract, you stated that:

"…upstream non-equilibrium sediment supply condition in poorly sorted sediment has a non-negligible role in downstream alternate bar dynamics **EVEN FAR FROM** the sediment point. However, this effect becomes negligible in the **FURTHER DOWNSTREAM REACHES**…". The terms "even far from" and "further downstream" are rather ambiguous to me.

2. In L174 of P6 you said that the model applied an active layer, a deposition layer, and a transition layer. In L175 of P6 you said that there are several deposition layers. Whether you applied one or multiple deposition layers? Please keep consistent in the text.

3. L200-201 P7: How do you calculate the flow velocity near the riverbed? What values do you specify for $\mu_s$ and $\mu_k$?

4. Fig. 2: Does the figure show water depth and bed elevation (as written in the caption), or the change in water depth and bed elevation from the initial bed to the end of the simulation (as written in L230 P8)?

5. L283-284 P12: If I understand correctly, do you apply a longer computational length for the wide channel cases than the narrow channel cases? The wide channel has 600 cells in the longitudinal direction while the narrow channel has 200 cells.

6. Fig. 7: Is this longitudinal riverbed averaged over the cross section?

7. Caption of Fig. 8: What do you mean by "upper low" and "lower low"?

8. Around L330 P15: In Fig. 8 you use the terms "upstream reach" and "downstream reach", but in the text you also use the term "middle reach". I think it would be good to keep consistent in the text.

9. L343 P17: A verb seems to be missing between "which" and "a".

10. L355 P18: How do you determine whether bedload sheets exist of dissipate from the plot? To me, the sediment transport rate still shows spatial variation at 0.5 $T_h$ in the grey area.

11. L369 P19: Why does the disappearance of bedload sheets leads to a counterclockwise hysteresis?

12. L392 P19: What distribution? Grain size distribution?

13. Fig. 13: All the three locations shown in the figure are beyond HBL. I am quite interested about the hysteresis pattern within the HBL.

14. L414-422 P21: If I understand correctly, previous papers cited here studied transient process due to the long scale change in sediment supply, whereas your research focuses on the dynamic equilibrium under time varying cyclic sediment supply.

15. L452-454 P22: Another mechanism recently found to be important for magic sand effect is the change of flow regime in the viscous sublayer. The authors might be interested to read the following two papers:

Parker G., An C., Lamb M. P., Garcia M. H., Dingle E. H., Venditti J. G. 2024. Dimensionless argument: a narrow grain size range near 2 mm plays a special role in river sediment transport and morphodynamics. Earth Surface Dynamics, 12, 367–380.

Hassan M. A., Parker G., Hassan Y., An C., Fu X., Venditti J. G. 2024. The roles of geometry and viscosity in the mobilization of coarse sediment by finer sediment. Proceedings of the National Academy of Sciences of the United States of America, 121(38): e2409436121.

16. L483-484 P23: Do you mean that a narrow grain size distribution causes the migration of bedload sheets?

---

## Author Response (AR1)

Thank you for giving us the opportunity to strengthen our manuscript with your valuable comments and queries. We have worked hard to incorporate your feedback. The following is our responses to reviewer's comments and detailed point-by-point response. Please note that the relevant figures are presented at the bottom of this letter, and supplement movies are attached.

Soichi Tanabe and Toshiki Iwasaki

Referee: Chenge An
[Comments to the Author]
It has been known that with cycled hydrographs, the alluvial river can reach a dynamic equilibrium under which only a limited distance from the inlet is influenced by the varying inflow discharge. This region is called a hydrograph boundary layer (HBL), and can be break down by advective bedload sheets when sediment is poorly sorted. However, previous studies only apply a 1D analysis, and cannot consider the alternate bars that are commonly observed in natural river. In this study, the authors extend previous studies to a 2D framework, and can therefore study the interaction of bedload sheets and alternate bars with the breakdown of HBL. That said, I regard this manuscript as a big progress on the knowledge of the HBL as well as the dynamic equilibrium of alluvial rivers. The topic is suitable for Esurf. Generally, the manuscript is well organized and easy to read. However, I still find a few issues that need to be addressed, with the major issue being about the governing equation. I list my detailed comments below.

[Author Response]
We appreciate your insightful comments. As you pointed out, we have corrected the governing equation (Equation 7) in the revised manuscript. Please note that the correct equations were already used in our numerical model, so the calculation results remain unaffected. In addition, we have added explanation of hysteresis in sediment transport rate within HBL. Please note that line numbers differ between the new manuscript (no track-changes) and the author's track-changes manuscript. For the "**Location in new manuscript**", we refer to the line numbers in the new manuscript (no track-changes).

<Main Comment>
**Location**: line 160
**Comment**: Equation (7) seems to be incorrect to me. Following is the equation from Chapter 4 of Parker's e-book (Parker, 2004). A major difference between your Eq. (7) and the following equation from Parker's e-book is that, Eq. (7) does not have fIi which represent the grain size distribution of the sediment exchanging between the active layer and the substrate material.

Besides, I am also confused with the L167-168 of Page 6. Doesn't Fai denote the grain size distribution of the active layer by definition?

**Answer**: Thank you for your comment. Equation 7 in the original manuscript was incorrect, and we have corrected it as the referee pointed out:

$$(1-\lambda)\left(f_{Ii}\frac{\partial}{\partial t}(\eta - L_a) + \frac{\partial}{\partial t}(F_{ai}L_a)\right) = -\left(\frac{\partial q_{Bi}{}^x}{\partial x} + \frac{\partial q_{Bi}{}^y}{\partial y}\right)$$

With respect to fIi, the grain size fraction in the active layer is adopted when the riverbed aggrades, and (i.e., the transition layer) is adopted when the riverbed degrades. Please note that the correct equations were already used in our numerical model, so we would like to emphasize that the calculation results are reliable.

**Location in new manuscript**: Lines 166-169

<Specific Comments>

**Location**: Lines 20-24 (Abstract)

**Comment**: In the last two sentences of the abstract, you stated that: "…upstream non-equilibrium sediment supply condition in poorly sorted sediment has a non-negligible role in downstream alternate bar dynamics **EVEN FAR FROM** the sediment point. However, this effect becomes negligible in the **FURTHER DOWNSTREAM REACHES**…". The terms "even far from" and "further downstream" are rather ambiguous to me.

**Answer**: We have removed "even far from the sediment feed point" in order to emphasize "further downstream" more clearly.

**Location in new manuscript**: Line 20

**Location**: Line 174

**Comment**: In L174 of P6 you said that the model applied an active layer, a deposition layer, and a transition layer. In L175 of P6 you said that there are several deposition layers. Whether you applied one or multiple deposition layers? Please keep consistent in the text.

**Answer**: We used multiple deposition layers to store the grain size distribution in substrate. To explain this, we have changed "a several deposition layer" to "several deposition layers".

**Location in new manuscript**: Line 160

**Location**: Lines 200-201

**Comment**: How do you calculate the flow velocity near the riverbed? What values do you specify for μs and μk?

**Answer**: We assumed that μs and μk have same value 0.7, corresponding to an angle of repose of 35 degrees. In addition, to calculate the flow velocity near the riverbed, we used the equilibrium secondary flow model proposed by Engelund (1974). We have added these descriptions.

**Location in new manuscript**: Lines 199-206

**Location**: Fig. 2
**Comment**: Does the figure show water depth and bed elevation (as written in the caption), or the change in water depth and bed elevation from the initial bed to the end of the simulation (as written in L230 P8)?
**Answer**: Thank you for your comment. These figures show water depth and detrended riverbed elevation at the equilibrium state (i.e., approximately 20 hours). We have added an explanation that the detrended riverbed elevation was obtained by subtracting the channel slope of 0.013 from the riverbed elevation at the equilibrium state.
**Location in new manuscript**: Fig. 2, Lines 235-236

**Location**: Lines 283-284
**Comment**: If I understand correctly, do you apply a longer computational length for the wide channel cases than the narrow channel cases? The wide channel has 600 cells in the longitudinal direction while the narrow channel has 200 cells.
**Answer**: Thank you for your comment. In fact, both the wide channel (bar cases) and the narrow channel (non-bar cases) use the same channel length (21 km). However, only grid size differs between them. The narrow channel has coarse grid because of the reduction in the computation time, but this is still sufficient to resolve the bedload sheet migration. To improve clarity, we have also revised the manuscript.
**Location in new manuscript**: Lines 264-265, 290-297

**Location**: Fig. 7
**Comment**: Is this longitudinal riverbed averaged over the cross section?
**Answer**: This is the longitudinal riverbed elevation along the right bank (Y = 0 m). To avoid confusion, we have added the label "Y=0 m" to Fig. 7 and manuscript as well as to the other relevant figures (Figs. 5 and 10).
**Location in new manuscript**: Figs. 5, 7 and 10, Lines 301, 324, 343

**Location**: Caption of Fig. 8
**Comment**: What do you mean by "upper low" and "lower low"?
**Answer**: This is mistake, our intension is "row" not "low". To make the description clearer and more natural, we have used "panel" instead.
**Location in new manuscript**: Figs. 8 and 9

**Location**: Lines 330
**Comment**: In Fig. 8 you use the terms "upstream reach" and "downstream reach", but in the

text you also use the term "middle reach". I think it would be good to keep consistent in the text.

**Answer**: We have used upstream (3-5km) and middle (10-12km) reaches to explain the computational result here.

**Location in new manuscript**: Line 328

**Location**: Lines 343

**Comment**: A verb seems to be missing between "which" and "a".

**Answer**: Thank you for your comment. This should be "which is a". We have corrected it.

**Location in new manuscript**: Line 349

**Location**: Lines 355

**Comment**: How do you determine whether bedload sheets exist of dissipate from the plot? To me, the sediment transport rate still shows spatial variation at 0.5 Th in the grey area.

**Answer**: As you pointed out, the effect of bedload sheets does not completely disappear at 0.5Th. Although no local peak in sediment transport volume is observed at 0.3Th and 0.5Th, spatial change of sediment transport rate within the gray area shows slight irregularity compared to that in the further downstream reach (Fig. 11). These irregularities are caused by the migration of bedload sheets.

**Location in new manuscript**: Lines 355-358

**Location**: Lines 369

**Comment**: Why does the disappearance of bedload sheets leads to a counterclockwise hysteresis?

**Answer**: Thank you for your comment. First, we thought that weak CCW hysteresis was observed outside the affected length of bedload sheets. However, we have finally concluded that this is negligible hysteresis compared to the one observed in HBL. Therefore, we have revised our interpretation and now consider that hysteresis is absent outside both HBL and the affected length of bedload sheets.

**Location in new manuscript**: Lines 370-388

**Location**: Lines 392

**Comment**: What distribution? Grain size distribution?

**Answer**: That is correct. We have added "grain size" for clarity.

**Location in new manuscript**: Line 403

**Location**: Fig. 13

**Comment**: All the three locations shown in the figure are beyond HBL. I am quite interested

about the hysteresis pattern within the HBL.

**Answer**: We have provided the hysteresis patterns at three locations: within HBL, within the affected length of bedload sheets, outside the affected length of bedload sheets in bar cases. In response to comment from other referee, we have also added the hysteresis patterns of riverbed variation and geometric mean diameter. Within the HBL, the strong CW hysteresis is observed in sediment transport volume. In other words, sediment transport volume at the rising limb is larger than at the falling limb. This is due to a larger riverbed slope at the rising limb compared to falling limb. In contrast, there is no obvious hysteresis in geometric mean diameter except for bedload sheets. It means that the hysteresis in riverbed variation is a key factor controlling the sediment transport hysteresis within the HBL. On the other hand, we have also found that, outside the HBL (i.e., X=3570m), the sediment transport hysteresis is still observed in Case 1 (i.e. more poorly sorted case). In this case, the magnitude of the riverbed variation is very small, but the geometric mean grain size shows strong hysteresis as shown in Fig. 15. It suggests that the sediment transport hysteresis observed outside the HBL is caused by the migration of bedload sheets.

**Location in new manuscript**: Figs. 13, 14 and 15, Lines 370-388

**Location**: Lines 414-422

**Comment**: If I understand correctly, previous papers cited here studied transient process due to the long scale change in sediment supply, whereas your research focuses on the dynamic equilibrium under time varying cyclic sediment supply.

**Answer**: As you pointed out, previous studies have focused on long-term changes in sediment supply. We aim to clarify how long the impact of cycled hydrograph and constant sediment supply, representing nonequilibrium sediment supply within the short term such as a single flood, propagates within alternate bars. We believed that our study helps distinguish between short- and long-term effects of changes in sediment supply condition on river morphodynamics.

**Location in new manuscript**: Lines 426-436

**Location**: Lines 452-454

**Comment**: Another mechanism recently found to be important for magic sand effect is the change of flow regime in the viscous sublayer. The authors might be interested to read the following two papers: Parker et al., 2024, Hassan et al., 2024

**Answer**: We appreciate your valuable information. In addition to the geometric mechanism previously described, we have added an explanation of the viscous mechanism.

**Location in new manuscript**: Lines 455-459

**Location**: Lines 483-484

**Comment**: Do you mean that a narrow grain size distribution causes the migration of bedload

sheets?

**Answer**: As this sentence did not correctly explain our original intention, we have revised it as follow: Original text: "This change in bar characteristics is unclear in the case of a narrower grain size distribution range, which causes the migration of bedload sheets; however, its intensity is much weaker." Revised text: "This effect of bedload sheets on bar morphology in the case with a narrower grain size distribution is weaker than that in the case with a wider distribution, owing to the smaller magnitude of bedload sheets."
**Location in new manuscript**: Lines 488-489

Referee: Anonymous Referee #2
[Comments to the Author]
This paper presents two-dimensional morphodynamic modeling simulations investigating how the so-called hydrograph boundary layer influences morphodynamics in straight channels with migrating alternate bars. This topic is a natural progression from prior work on hydrograph boundary layers, which have focused on the one-dimensional (bar-free) condition up until now. I therefore think this paper is relevant and of potential interest to the readers of Earth-Surface Dynamics.
The research presented in the paper is nicely conceived and generally well presented. There are some areas of the presentation and analysis that I find somewhat confusing, and I encourage the authors to consider the comments below in revision.

[Author Response]
We appreciate your valuable comments and suggestions. First, we have revised the description of the initial conditions to improve clarity. As you recommended, we have included movies to enhance the understanding our results. Furthermore, we have added figures illustrating hysteresis of riverbed elevation and geometric mean diameter, as well as observations regarding hysteresis within the HBL. Please note that line numbers differ between the new manuscript (no track-changes) and the author's track-changes manuscript. For the "**Location in new manuscript**", we refer to the line numbers in the new manuscript (no track-changes).

<Comments>
**Location**: Chapter 3.1
**Comment**: More details on the specific set up of the model are needed. What are the initial conditions of the simulations? Was an initial equilibrium bed developed before imposing the hydrograph and constant sediment supply? What was the time step and cell size?
**Answer**: The initial riverbed geometry for all cases, including model validation, was a flatbed, and we have added this information. We did not conduct preliminary simulation prior to the cycled hydrograph. We have added the time step and cell size.

**Location in new manuscript**: Lines 264-265, 295-297

**Location**: Lines 285-290

**Comment**: The description of the calculation of sediment supply rates is also unclear.

**Answer**: After determining the hydrograph, channel slope and grain size distribution, we selected sediment supply rate through a trial-and-error process to satisfy macro-scale equilibrium during a single hydrograph. We have improved the explanation as follow:

Original text: "The constant sediment supply rate in the simulation is determined by a combination of the channel slope, hydrograph, and sediment size distribution. In the simulations, we fix the hydrograph, channel slope, and sediment size distribution and adjust the sediment supply rate from the upstream end to achieve macro-scale morphodynamic equilibrium, i.e. the only variation during the hydrograph is upstream bed fluctuation and migration of the grain-sorting wave, while the macroscale bed slope is maintained." Revised text: "The constant sediment supply rate in the simulation was determined through a trial-and-error approach, because there is no straightforward, explicit method. The channel slope, hydrograph shape, and grain size distribution all determine the constant sediment supply rate required to achieve macroscale dynamic equilibrium over a single hydrograph (i.e. the only variations during the hydrograph are upstream bed fluctuation and migration of the grain-sorting wave, while the macroscale bed slope is maintained)."

**Location in new manuscript**: Lines 284-288

**Location**: Fig. 5 and others

**Comment**: For Fig 5 (and others) – is it possible to provide movie files of the simulation results as supplemental information? It would be informative to be able to see the evolution of these simulations as they occur.

**Answer**: We appreciate your valuable advice. We have added movies for Figs. 5, 7, 11 and 12.

**Location in new manuscript**: Figs. 5, 7, 11 and 12, Movies S1-S8

**Location**: Chapter 2.2 and 3.1

**Comment**: Can you provide more description of how the bars emerged? No mention is made of any initial topographic instability, so what led to bar emergence and why at 3km?

**Answer**: We apologize for the lack of the information. A 5 % discharge fluctuation was randomly distributed in transverse direction at the upstream end for a perturbation to trigger bar formation. The above information has been added. Moreover, the additional movies (Movies S3 and S4) allow us to observe the process of bar formation and migration. We hope these movies help understand the computational result.

**Location in new manuscript**: Lines 265-267, Movies S3 and S4

**Location**: Fig. 8

**Comment**: What is "upper low" and "lower low"?

**Answer**: This is mistake, our intension is "row" not "low". To make the description clearer and more natural, we have used "panel" instead.

**Location in new manuscript**: Figs. 8 and 9

**Location**: Fig. 13

**Comment**: Figure 13 is somewhat challenging to understand. What does the hysteresis look further upstream (within the HBL)? I wonder if also showing the hysteresis of bed elevation deviation, and/or average surface geometric mean grain size, at the locations chosen for Figure 13 would more clearly show the impact of bedload sheets.

**Answer**: Fig. 13 (14 and 15) presents the temporal variation in flow discharge and sediment transport volume (riverbed variation and geometric mean diameter, respectively). This type of figure helps to understand the hysteresis pattern. To improve the clarity of figures, we have added time labels for specific times (0Th and 0.5Th) to the top-left panel. As you pointed out, we have also added the hysteresis patterns of riverbed variation and geometric mean diameter. We appreciate your insightful advice, as these additional figures are useful for understanding and discussing both the hysteresis pattern and the behaviour of bedload sheets. Within the HBL, the strong CW hysteresis is observed in sediment transport volume. In other words, sediment transport volume at the rising limb is larger than at the falling limb. This is due to a larger riverbed slope at the rising limb compared to falling limb. In contrast, there is no obvious hysteresis in geometric mean diameter except for bedload sheets. It means that the hysteresis in riverbed variation is a key factor controlling the sediment transport hysteresis within the HBL. On the other hand, we have also found that, outside the HBL (i.e., X=3570m), the sediment transport hysteresis is still observed in Case 1 (i.e. more poorly sorted case). In this case, the magnitude of the riverbed variation is very small, but the geometric mean grain size shows strong hysteresis as shown in Fig. 15. It suggests that the sediment transport hysteresis observed outside the HBL is caused by the migration of bedload sheets.

**Location in new manuscript**: Figs. 13, 14 and 15, Lines 370-388

---

## Author Response (AR3)

Thank you for accepting our manuscript. We have addressed the comments regarding the colour scheme from the editorial support team as much as possible. The table and the caption for the figures and tables are attached at the end of the text file.

Soichi Tanabe and Toshiki Iwasaki